# Review of Constituents and Biological Activities of Triterpene Saponins from *Glycyrrhizae Radix* et Rhizoma and Its Solubilization Characteristics

**DOI:** 10.3390/molecules25173904

**Published:** 2020-08-27

**Authors:** Feifei Li, Bin Liu, Tong Li, Qianwen Wu, Zhiyong Xu, Yuhao Gu, Wen Li, Penglong Wang, Tao Ma, Haimin Lei

**Affiliations:** 1School of Chinese Pharmacy, Beijing University of Chinese Medicine, Beijing 100102, China; lifeifei902@163.com (F.L.); lt1755258545@163.com (T.L.); winifredG@163.com (Q.W.); gu2452260684@163.com (Y.G.); lw19991103@163.com (W.L.); wangpenglong@bucm.edu.cn (P.W.); mosesmatao@163.com (T.M.); 2Institute of Regulatory Science for Traditional Chinese Medicine, Beijing University of Chinese Medicine, Beijing 100102, China; 3COFCO Nutrition and Health Research Institute, Beijing 102209, China; liubin11@cofco.com; 4Key Laboratory of Computational Chemistry-Based Natural Antitumor Drug Research & Development, School of Traditional Chinese Materia Medica, Shenyang Pharmaceutical University, Shenyang 110016, China; XZY9525@163.com

**Keywords:** *Glycyrrhizae Radix* et Rhizoma, triterpene saponins, glycyrrhizin, biological activities, solubilization

## Abstract

*Glycyrrhizae Radix* et Rhizoma is regarded as one of the most popular and commonly used herbal medicines and has been used in traditional Chinese medicine (TCM) prescriptions for over 2000 years. Pentacyclic triterpene saponins are common secondary metabolites in these plants, which are synthesized via the isoprenoid pathway to produce a hydrophobic triterpenoid aglycone containing a hydrophilic sugar chain. This paper systematically summarizes the chemical structures of triterpene saponins in *Glycyrrhizae Radix* et Rhizoma and reviews and updates their main biological activities studies. Furthermore, the solubilization characteristics, influences, and mechanisms of *Glycyrrhizae Radix* et Rhizoma are elaborated. Solubilization of the triterpene saponins from *Glycyrrhizae Radix* et Rhizoma occurs because they contain the nonpolar sapogenin and water-soluble sidechain. The possible factors affecting the solubilization of *Glycyrrhizae Radix* et Rhizoma are mainly other crude drugs and the pH of the decoction. Triterpene saponins represented by glycyrrhizin from *Glycyrrhizae Radix* et Rhizoma characteristically form micelles due to amphiphilicity, which makes solubilization possible. This overview provides guidance regarding a better understanding of *Glycyrrhizae*
*Radix* et Rhizoma and its TCM compatibility, alongside a theoretical basis for the further development and utilization of *Glycyrrhizae Radix* et Rhizoma.

## 1. Introduction

The *Glycyrrhiza* genus belongs to the Fabaceae family, comprising approximately 20 species primarily distributed across Asia, Europe, North America, and South America, with eight distributed throughout China [1]. *Glycyrrhizae Radix* et Rhizoma, also named glycyrrhiza or “Gan-Cao” in China and licorice or liquorice in Europe, is the dry root and rhizome of three official *Glycyrrhiza* species, namely, *Glycyrrhiza uralensis* Fisch, *Glycyrrhiza glabra* L., and *Glycyrrhiza inflata* Batal [2,3]. It is extensively used in traditional Chinese medicine (TCM) to treat hepatitis, influenza, cough, and gastric ulcers [4,5]. Glycyrrhiza is also of significant economic value, and its extract has been used in cosmetics, food ingredients, tobacco flavors, and functional foods [5,6,7,8,9].

In recent decades, extensive research has been conducted regarding the bioactive constituents, biosynthesis, pharmacological mechanisms, and clinical applications, in glycyrrhiza, among other aspects [5,10,11,12,13]. The major bioactive secondary metabolites of glycyrrhiza include triterpene saponins, various types of flavonoids, coumarins, polysaccharides, and other phenolics [10,14]. Unfortunately, no systematic review has been conducted as yet regarding the chemical structure, origin, and corresponding references of triterpenoid saponins. Even the number of triterpenoid saponins in some references is not up-to-date. Another aspect of concern is that the solubilization characteristics of glycyrrhiza also received increasing attention in recent years due to the possibility of triterpene saponins from glycyrrhiza increasing the solubility of coexisting bioactive constituents in herbal extracts [15]. Few reviews currently exist regarding this subject.

In this review, SciFinder, PubMed, Web of Science, China Journal Net, and relevant English and Chinese literature were used as information sources by the inclusion of the primary search terms “*Glycyrrhizae Radix*”, “glycyrrhiza”, “Gan-Cao", “liquorice”, “licorice”, “triterpene saponins”, “constituent”, “glycyrrhizin”, “glycyrrhizic acid”, “activities”, “solubilization”, and their combinations, mainly from 1984 to 2020. We systematically summarize the chemical structures, origins, and solubilization characteristics of triterpene saponins in glycyrrhiza and mainly focus on their chemical structures and characterization as natural surfactants. In addition, their biological activities are also reviewed and updated.

## 2. Triterpene Saponins and Their Bioactivities

### 2.1. Triterpene Saponins

The investigations of the chemical constituents of glycyrrhiza led to the isolation of 77 triterpene saponins. Triterpenoid saponins are major components of glycyrrhiza, containing one or more sugar moieties attached to oleanane-type pentacyclic triterpenoid aglycones. All of the triterpenoid saponins (Figure 1, Figure 2 and Figure 3) in glycyrrhiza are summarized in Table 1. There were 50 oleanane-type pentacyclic triterpene saponins obtained from *G. uralensis* (Figure 1), 38 from *G. glabra* (Figure 2), and only 13 from *G. inflate* (Figure 3).

From the point of view of chemical structure, the aglycons of most oleanane-type pentacyclic triterpene saponins in glycyrrhiza possess an α,β-unsaturated ketone unit located at C-11, C-12, and C-13. Notably, some glabrolides (**7**, **21**, **22**, **30**, **39**, **42**, and **46**) were found in *G. uralensis* or/and *G. inflate*, which possess a 22 (30)-lactone ring alongside an α,β-unsaturated ketone unit. The sugar moiety of oleanane-type pentacyclic triterpene saponins in glycyrrhiza contains six basic sugar residues, including glucuronic acid residue (GluA), rhamnose residue (Rha), glucose residue (Glu), galacturonic acid residue (GalA), xylose residue (Xyl), and galactose residue (Gal). Moreover, apioglycyrrhizin (**15**) contains an apiofuranose residue and araboglycyrrhizin (**16**) contains an arabinose residue (Ara). Furthermore, all triterpene saponins in glycyrrhiza are linked to sugar groups at C-3; the glycoside bound to C-3 of the aglycon possesses the β-configuration. Among them, the C-21 linked to glycoside is the β-configuration (**73**–**76**), while the hydroxyl group at C-21 has both the α-configuration and β-configuration.

Glycyrrhizin (GL, **1**) (also named glycyrrhizic acid, uralsaponin A, and 18β-glycyrrhizic acid) is one of the most representative saponins of glycyrrhiza, isolated from the roots of both *G. uralensis* Fisch. [16], *G. glabra* L. [19], and *G. inflata* Batal. [20]. Zapesochnaya et al. [37] demonstrated the differences between the NMR spectra of the 18α-epimer of GL and the 18β-epimer. Normally, 18β-glycyrrhizic acid is the principal chemical composition, while 18α-glycyrrhizic acid is rare [38]. Licorice-saponin Q2 (**44**) was previously isolated from the roots of *G. inflata.* Analysis of its Nuclear Overhauser Effect Spectroscopy (NOESY) spectrum showed that H-18 correlated with H-19α and H-29(CH_3_), indicating that H-18 of **44** was α-oriented. In addition, similar triterpene saponins, such as licorice-saponin G2 (**9**) and araboglycyrrhizin (**16**), were also observed in *G. uralensis*, *G. glabra,* and *G. inflate*. Eleven triterpene saponins (**2**, **4**, **5**, **10**–**12**, **14**, **25**, **26**, **36**, and **41**) were also found both in *G. uralensis* and *G. glabra*, including uralsaponin B (**2**) [16,21], licorice-saponin B2 (**4**) [22,23], licorice-saponin C2 (**5**) [22,23], licorice-saponin H2 (**10**) [23,24], licorice-saponin J2 (**11**) [23,24], licorice-saponin K2 (**12**) [21,24], 18α-glycyrrhizic acid (**14**) [17], licorice-saponin M3 or uralsaponin T (**25**) [27,31], licorice-saponin N4 (**26**) [31,32], uralsaponin V (**36**) [21,27], and 3-*O*-β-d-glucuronopyranosylglycyrrhetinic acid (**41**) [21,27]. Licorice-saponin M3, and uralsaponin T were previously reported as new oleanane-type triterpene saponins, but they are the same compounds.

### 2.2. Biological Activities

Modern pharmacological studies revealed that glycyrrhiza shows a variety of pharmacological effects against inflammation, oxidative stress, immunoregulation, viral infection, and cancer [3]. These bioactivities are attributed to the chemical constituents of glycyrrhiza. In this section, the main pharmacological activities of saponin monomers, including hepatoprotective, anti-inflammatory, antimicrobial, antiviral, and antitumor activities, are summarized (Table 2). GL (**1**) is the most commonly reported monomer with extensive activities.

#### 2.2.1. Hepatoprotective Activities

There are many reports about possible mechanisms in vitro and vivo by which saponins from glycyrrhiza are hepatoprotective. GL (**1**) was proven to relieve liver disease and prevent drug-induced liver injury through multitargeting therapeutic mechanisms, including antisteatosis, antioxidative stress, anti-inflammation, immunoregulation, antifibrosis, anticancer, and drug–drug interactions [3]. Nakamura et al. [39] reported that GL (**1**) prevented soluble enzyme release from primary cultured rat hepatocytes induced by CCl_4_. Sato et al. [40] found that GL (**1**) could modify the expression of hepatitis B virus (HBV)-related antigens on the hepatocytes and suppress sialylation of hepatitis B surface antigen (HBsAg) in PLC/PRF/5 cells. Tsuruoka et al. [41] showed that GL (**1**, 10.5 mg/kg) suppressed increases in aspartate aminotransaminase (AST) and alanine aminotransaminase (ALT), inhibited inducible nitric oxide synthase (iNOS) mRNA expression, and reduced protein and cell infiltration and the degeneration of hepatocytes in the liver of concanavalin A (Con A)-treated BALB/c mice. Lee et al. [42] reported that GL (**1**) alleviated carbon tetrachloride (CCl_4_)-induced liver injury in ICR mice, probably by inducing heme oxygenase-1 and downregulating proinflammatory mediators. Lin et al. [43] found that a three-day pretreatment with GL (**1**) exhibited a protective effect on retrorsine-induced liver damage in Sprague Dawley rats. GL (**1**) is able to provide partial protection of the liver against ischemia-reperfusion damage in Wistar rats [44]. Orazizadeh et al. [45] showed that GL (**1**) effectively protects against NTiO_2_-induced hepatotoxicity in Wistar rats.

In addition, some other triterpenoid saponins in glycyrrhiza also exhibited hepatoprotective activities. Glyuralsaponin B (**64**) and glyuralsaponin H (**70**) exhibited moderate antioxidant activities against Fe^2+^/cysteine-induced liver microsomal lipid peroxidation at a concentration of 0.1 μM (curcumin as positive control) [32]. It was reported that GL (**1**), licorice-saponin G2 (**9**), 22β-acetoxylglycyrrhizin (**17**), licorice-saponin Q2 (**44**), and macedonoside A (**45**) showed significant hepatoprotective activities by lowering ALT and AST levels in primary rat hepatocytes injured by D-galactosamine (D-GalN) in a concentration range of 30–120 μM. Besides, GL (**1**), licorice-saponin G2 (**9**), 22β-acetoxylglycyrrhizin (**17**), uralsaponin D (**21**), licorice-saponin Q2 (**44**), and macedonoside A (**45**) were found to potently inhibit the activity of phospholipase A2 (PLA_2_) with IC_50_ values of 9.3 μM, 16.9 μM, 27.1 μM, 32.2 μM, 3.6 μM, and 6.9 μM, respectively, which might be involved in the regulation of the hepatoprotective activities observed. [20].

#### 2.2.2. Anti-Inflammatory Activities

In a study, Li et al. [46] suggested that the anti-inflammatory mechanism of total saponins of glycyrrhiza may be related to a reduction in the release of inflammation factors in macrophages and inhibition of the key enzymes in the arachidonic acid (AA) metabolism pathway of prostaglandin E2 (PGE_2_) synthesis, as observed through an inflammatory model of mouse macrophage RAW264.7 cells induced by lipopolysaccharide (LPS). Wang et al. [47] investigated the anti-inflammatory effect of GL (**1**) on LPS-stimulated mouse endometrial epithelial cells (MEEC), demonstrating that GL (**1**) inhibited LPS-induced inflammatory response by inhibiting TLR4 signaling pathway in MEEC. Akamatsu et al. [48] found that GL (**1**) inhibited reactive oxygen species (ROS) generation by neutrophils, which were potent inflammatory mediators in the in vitro study.

In addition, GL (**1**) may inhibit high-mobility group protein B1 (HMGB1) expression and subsequent production of inflammatory cytokines to prevent cerebral vasospasm (CVS) following subarachnoid hemorrhage (SAH) in Sprague-Dawley rats [49]. Pang et al. [50] demonstrated that inhibiting HMGB1 with GL (**1**) alleviated brain injury after diffuse axonal injury (DAI) via its anti-inflammatory effects in SD rats.

#### 2.2.3. Antimicrobial and Antiviral Activities

Saponins of *G. glabra* L. have broad-spectrum antimicrobial activities and can be used as natural antimicrobial agents [51]. GL (**1**) is an effective antiviral component against hepatitis C virus (HCV), human immunodeficiency virus (HIV), coxsackie virus B3 (CVB3), duck hepatitis virus (DHV), enterovirus 71 (EV71), coxsackievirus A16 (CVA16), herpes simplex virus (HSV), and H5N1 by weakening viral activity and enhancing host cell activity [52]. GL (**1**) is also shown to inhibit varicella zoster virus (VZV) and the severe acute respiratory syndrome coronavirus (SARS-CoV) replication in vitro [53,54]. In another study, Wolkerstorfer et al. [55] found that GL (**1**) inhibited influenza A virus (IAV) uptake into the cell. In detail, Sun et al. [56] summarized the antiviral effects of GL (**1**) in their research regarding progress and mechanism in recent years.

At present, the world is facing the Corona Virus Disease 2019 (COVID-19) pandemic, caused by severe acute respiratory syndrome coronavirus-2 (SARS-CoV-2). GL (**1**) has been used to control COVID-19 infections, which may reduce the severity of an infection with COVID-19 at the two stages of the COVID-19-induced disease process: 1. to block the number of entry points and 2. to provide an angiotensin converting enzyme 2 (ACE2)-independent anti-inflammatory mechanism. [57]. In vitro assays of 22β-acetoxyglycyrrhizin (**17**), uralsaponin T (**25**), uralsaponin M (**28**), and uralsaponin S (**34**) exhibited good inhibitory activities against influenza virus A/WSN/33 (H1N1) in Madin–Darby canine kidney (MDCK) cells (using Oseltamivir phosphate as a positive control drug) [27]. In addition, GA (**1**), licorice-saponin A3 (**3**), licorice-saponin G2 (**9**), 22β-acetoxylglycyrrhizin (**17**), and licorice-saponin M3 (**25**) were shown to possess moderate influenza neuraminidase (NA)-inhibitory activity by the commercial NA inhibitory screening kit, although the measured activity was lower than that of Oseltamivir [31].

#### 2.2.4. Cytotoxic and Antitumor Activities

Deng et al. [58] showed that GL (**1**) profoundly reduced expression of thromboxane synthase (TxAS), as well as proliferating cell nuclear antigen (PCNA), and rescued liver and kidney damage in tumor-bearing mice, the effect of which is possibly through suppression of the TxA2 pathway. It was shown that GL (**1**) has protective effects against Aflatoxin B1 (AFB_1_)-induced cytotoxicity in human hepatoma cell line (HepG2) [59]. In addition, dipotassium glycyrrhizinate (DPG), a dipotassium salt of GL, presented antitumoral effects on glioblastoma (GBM) cell lines through decreased proliferation and increased apoptosis. The DPG antitumoral effect is related to NF-*κ*B suppression, where *IRAK2*- and *TRAF6*-mediating *miR16* and *miR146a*, respectively, might be potential therapeutic targets of DPG [60].

In the cytotoxic assay, GL (**1**), licorice-saponin G2 (**9**) and uralsaponin D (**21**), showed no cytotoxic activity on tested cancer cell lines, whereas their corresponding aglycones exhibited potently cytotoxic activities against human cervical cancer HeLa cells and human breast adenocarcinoma MCF-7 cells [29].

#### 2.2.5. Other Activities

Saponin monomers of glycyrrhiza were shown to have various other physiological and pharmacological activities. GL (**1**) also possesses immunomodulatory, and neuroprotective effects [61] and antioxidant activities [62,63]. Furthermore GL (**1**) can be used in the clinical treatment of bronchitis, peptic ulcers, skin diseases, and oral diseases [56,64,65].

In addition, GA (**1**) may have a therapeutic effect on allergic rhinitis, partly by modulation of the Th1/Th2 balance through suppression of OX40 and by increasing the activity of regulatory T cells [66].

## 3. Solubilization Characteristics

In nature, saponins are distributed in 90 plant families from 500 genera [67]. Some of them have the potential to be used as natural surfactants because they contain the nonpolar sapogenin and water-soluble sidechain [68]. Glycyrrhiza is the most frequently used TCM in TCM formulae, with the function of harmonizing all kinds of TCMs. Research on the chemistry, pharmacological effects, clinical applications et al. of glycyrrhiza has been very extensive in recent decades. Besides, the saponins from glycyrrhiza have also significant solubilizing effects [69]. Interestingly, the solubilization characteristics of glycyrrhiza and saponins from glycyrrhiza were studied extensively over recent years. This part of review will deal with the solubilization characteristics, influences, and mechanisms regarding glycyrrhiza and triterpene saponins from glycyrrhiza (Table 3).

### 3.1. Solubilization Characteristics of Glycyrrhiza

Shi et al. [70] reported that glycyrrhiza has solubilization effects in TCM formulae, including sijunzi decoction, huangqi dazao decoction, and baishao gancao decoction, further explaining that the solubilizing components in glycyrrhiza are triterpene saponins. Meng et al. [71] studied and analyzed the decoctions of ephedra and glycyrrhiza, demonstrating that, compared to that of a single decoction, the contents of GL (**1**), ephedrine (including pseudoephedrine), and methephedrine (including methylpseudoephedrine) in the combined decoction of ephedra and glycyrrhiza were increased by 13.50%, 14.52%, and 64.0%, respectively. Nie et al. [72] demonstrated that after administration of a combined decoction of epimedium and glycyrrhiza, the contents of some chemical constituents, such as icariin in epimedium, were increased. Han et al. [73] reported that when extracted with 30% ethanol (*v*:*v*) with a 1:1 ratio of glycyrrhiza to curcuma longa, the extractive rate of curcumin doubled. At the same time, other studies also found that glycyrrhiza increased the contents of active ingredients in codonopsis, poria, atractylodes [74], *Baphicacanthus cusia* [75], *Paeoniae Radix Alba* [76], Isatidis Radix [77], and *Scutellaria baicalensis* [78].

Glycyrrhiza plays a significant role in solubilizing insoluble components, improving the bioavailability of active components and enhancing efficacy, reducing toxicity, and improving taste [79,80]. It should be noted that not all glycyrrhiza compounds in TCM formulae have surface activity, because solubilization is also affected by some crude drugs, such as *Schisandra chinensis* [70]. As for the mechanism of solubilization, most researchers demonstrated that the saponins of glycyrrhiza significantly reduced surface tension to play a solubilizing role, and further agreed that GL (**1**) is the main surfactant in glycyrrhiza [73,74,75,78,81,82].

### 3.2. Solubilization Characteristics of GL (***1***)

Sasaki et al. [83] found that the water solubility of saikosaponin-a, the active principle of Bupleurum root, is increased in the presence of water extract or the saponin fraction of glycyrrhiza and that this solubilizing effect is due to GL (**1**). Du [82] demonstrated that glycyrrhiza exhibits solubilization on Ben Lamge granules, thereby proving that GL (**1**) possesses solubilization activity. In detail, the solubility of Ben Lamge granules increases with the addition of GL (**1**), whereas the surface tension of GL (**1**) decreases. Experimental results also indicated that GL (**1**) exists in micelles in aqueous solution, where the critical micelle concentration (CMC) is 1.188 mg/mL. Lu et al. [84] showed that GL (**1**) exhibits certain solubilization on baicalin, and the dissolution rate of baicalin increases gradually as the concentration of GL (**1**) increases. The optimal CMC of GL (**1**) is 0.22 mg/mL regarding compatibility between *Scutellaria baicalensis* and glycyrrhiza, at which time the solubilization of GL (**1**) is at its highest. Yang et al. [85] reported that when the ratio of pueraria and glycyrrhiza was 5:3, the dissolution of puerarin reached its maximum and the CMC of GL (**1**) was 0.18 mg/mL, which was affected by the structure of the drug and the pH value of the solution. Cai et al. [69] demonstrated that GL (**1**) increased the solubility of pachymic acid in an aqueous solution, thereby improving the bioavailability of pachymic acid. Liu et al. [86] demonstrated that puerarin-glycyrrhizic acid dispersible tablets could improve the dissolution of puerarin in vitro due to the solubilization effect of GL (**1**).

GL (**1**) was reported to possess amphiphilic components consisting of one triterpenoid aglycone molecule and two glycosyl groups. With its inward hydrophobic group (triterpenoid aglycone) and outward hydrophilic group (two glucuronic acids), GL (**1**) spherical micelles can form in aqueous solution to increase the solubility of hydrophobic drugs. Petrova et al. [87] showed that NMR chemical shifts of the protons of the GL (**1**) glucuronic moiety were sensitive to solution pH and not sensitive to GL (**1**) concentration changes during GL (**1**) aggregation. At the same time, the protons of the triterpene moiety were shown to be sensitive to the nearest environment, and micelles formed via hydrophobic interaction between the triterpene moieties of GL (**1**).

GL (**1**) has potential applications as a biosurfactant in various fields [88]. In recent years, GL (**1**) was trialed for use as a “vehicle for drug delivery”, showing great potential in this field [89,90,91,92,93,94]. The solubilization effect of GL (**1**) is also widely used in the food industry [9,95].

## 4. Conclusions

This review provides an up-to-date summary concerning the phytochemistry and pharmacology of glycyrrhiza. Three species of the genus Glycyrrhiza—*G. uralensis*, *G. glabra*, and *G. inflata*—are considered to have a shared botanical origin of *Glycyrrhizae Radix* et Rhizoma. By the end of 2020, 77 triterpene saponins were discovered and identified from these plants. Previous phytochemical investigations revealed that triterpene saponins are one of the major constituents contributing either directly or indirectly to the biological effects of glycyrrhiza. Over recent decades, total saponins or saponin monomers from glycyrrhiza were found to possess various biological activities, such as hepatoprotective, anti-inflammatory, antimicrobial, antiviral, antitumor, antioxidant, and neuraminidase-inhibitory activities. Currently, these pharmacological studies on glycyrrhiza are limited to bioassays of only a few saponin monomers. Hence, further studies are needed to investigate the biological activities of more triterpene saponins via in vitro/vivo models. In addition, the phytochemistry of glycyrrhiza requires further study, with new compounds or degradation products possibly showing new pharmacological activities.

Another aim of this review was to summarize the solubilization characteristics, influences, and mechanisms regarding triterpene saponins from glycyrrhiza. Glycyrrhiza is widely used in TCM formulae and plays an important role in solubilizing insoluble components, thereby improving the bioavailability of active components, enhancing efficacy, and reducing toxicity, as well as demonstrating various pharmacological effects. The solubilization of glycyrrhiza with ephedra, epimedium, curcuma longa, codonopsis, poria, atractylode, *Baphicacanthus cusia*, *Paeoniae Radix Alba*, Isatidis Radix, and *Scutellaria baicalensis* was previously explored and the possible factors affecting the solubilization of glycyrrhiza were discussed, including some crude drugs and the pH of decoctions. Therefore, not all glycyrrhiza in a TCM formula possess solubilization activity. The solubilization of GL (**1**), the main solubilizing component from glycyrrhiza, with multiple active components from some other TCMs was also explored. In the discussion of solubilization mechanisms, the triterpene saponins represented by GL (**1**) from glycyrrhiza were shown to characteristically form micelles due to their amphiphilicity, thereby showing solubilization ability. The optimal CMC of micelle formation is different when GL (**1**) is combined with different TCMs or active ingredients. The pH of the solution was also shown to be critical to the formation of micelles. This review provides guidance regarding the better understanding of TCM compatibility and a theoretical basis for the further development and utilization of glycyrrhiza.

## Figures and Tables

**Figure 1 molecules-25-03904-f001:**
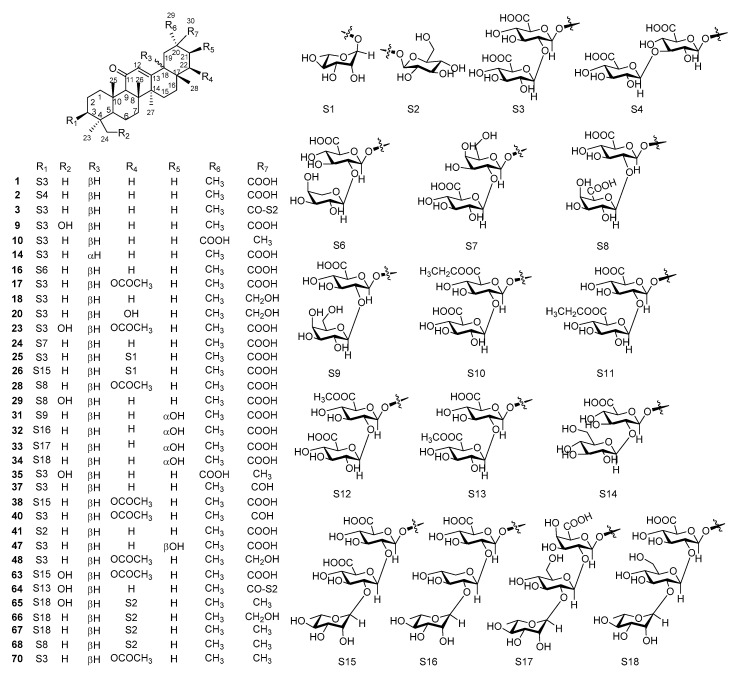
Triterpene saponins in *G. uralensis.*

**Figure 2 molecules-25-03904-f002:**
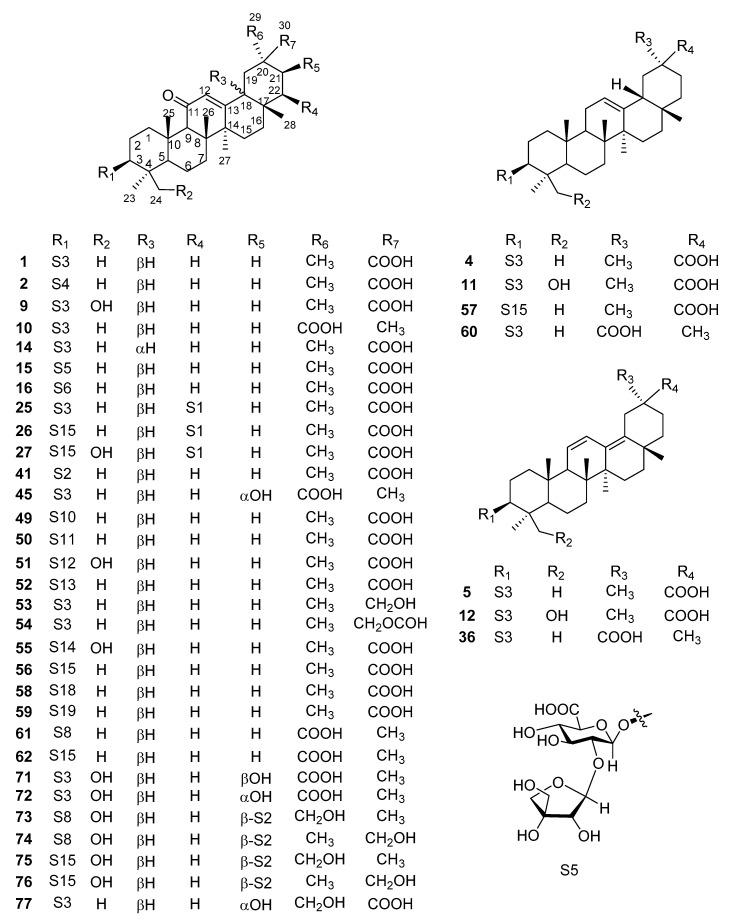
Triterpene saponins in *G. glabra.*

**Figure 3 molecules-25-03904-f003:**
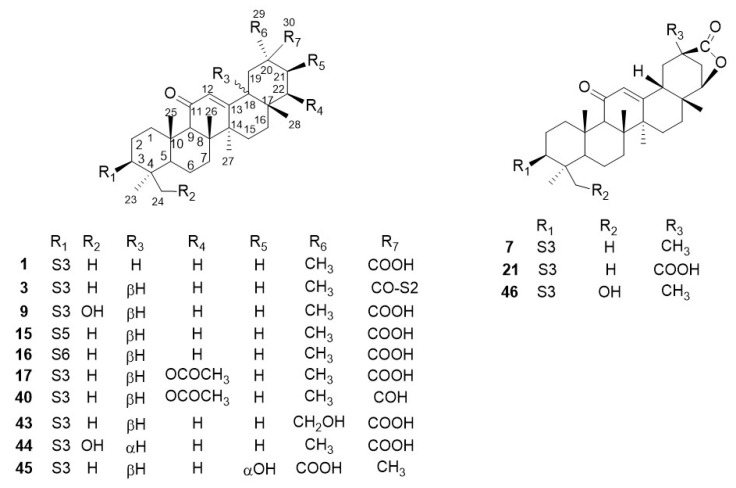
Triterpene saponins in *G. inflata.*

**Table 1 molecules-25-03904-t001:** Information on triterpene saponins in glycyrrhiza.

No.	Compound	Origin	References
**1**	glycyrrhizin (glycyrrhizic acid, uralsaponin A or 18β-glycyrrhizic acid)	a,b,c	[16,17,18,19,20]
**2**	uralsaponin B	a,b	[16,21]
**3**	licorice-saponin A3	a,c	[20,22]
**4**	licorice-saponin B2	a,b	[22,23]
**5**	licorice-saponin C2	a,b	[22,23]
**6**	licorice-saponin D3	a	[22]
**7**	licorice-saponin E2	a,c	[20,22]
**8**	licorice-saponin F3	a	[24]
**9**	licorice-saponin G2	a,b,c	[20,23,24]
**10**	licorice-saponin H2	a,b	[23,24]
**11**	licorice-saponin J2	a,b	[23,24]
**12**	licorice-saponin K2	a,b	[21,24]
**13**	licorice-saponin L3	a	[25]
**14**	18α-glycyrrhizic acid	a,b	[17]
**15**	apioglycyrrhizin	b,c	[23,26]
**16**	araboglycyrrhizin	a,b,c	[23,26,27]
**17**	22β-acetoxylglycyrrhizin	a,c	[20,28]
**18**	3β-*O*-[β-d-glucuronopyranosyl-(1→2)-β-d-glucuronopyranosyl]-glycyrretol	a	[28]
**19**	3β-*O*-[β-d-glucuronopyranosyl-(1→2)-β-d-glucuronopyranosyl]- olean-9,12-diene-30-oic acid	a	[28]
**20**	uralsaponin C	a	[29]
**21**	uralsaponin D	a,c	[20,29]
**22**	uralsaponin E	a	[29]
**23**	uralsaponin F	a	[29]
**24**	3-*O*-[β-d-glucuronopyranosyl-(1→2)-β-d-galactopyranosyl]glycyrrhetic acid	a	[30]
**25**	licorice-saponin M3 (uralsaponin T)	a,b	[27,31]
**26**	licorice-saponin N4	a,b	[31,32]
**27**	licorice-saponin O4	b	[31]
**28**	uralsaponin M	a	[27]
**29**	uralsaponin N	a	[27]
**30**	uralsaponin O	a	[27]
**31**	uralsaponin P	a	[27]
**32**	uralsaponin Q	a	[27]
**33**	uralsaponin R	a	[27]
**34**	uralsaponin S	a	[27]
**35**	uralsaponin U	a	[27]
**36**	uralsaponin V	a,b	[21,27]
**37**	uralsaponin W	a	[27]
**38**	uralsaponin X	a	[27]
**39**	uralsaponin Y	a	[27]
**40**	22β-acetoxyl-glycyrrhaldehyde	a,c	[20,33]
**41**	3-*O*-β-d-glucuronopyranosyl-glycyrrhetinic acid	a,b	[21,27]
**42**	3-*O*-[β-d-(6-methyl)glucuro-nopyranosyl (1→2)-d-glucurono-pyranosyl]-24-hydroxyglabrolide	a	[34]
**43**	licorice-saponin P2	c	[20]
**44**	licorice-saponin Q2	c	[20]
**45**	macedonoside A	b,c	[20,21]
**46**	24-hydroxy-licorice-saponin E2	c	[20]
**47**	macedonoside E	a	[35]
**48**	22β-acetyl-uralsaponin C	a	[35]
**49**	licorice saponin M1	b	[21]
**50**	licorice saponin M2	b	[21]
**51**	licorice saponin M3	b	[21]
**52**	licorice saponin M4	b	[21]
**53**	30-hydroxyglycyrrhizin	b	[23]
**54**	glycyrrhizin-20-methanoate	b	[23]
**55**	24-hydroxyglucoglycyrrhizin	b	[23]
**56**	rhaoglycyrrhizin	b	[23]
**57**	11-deoxorhaoglycyrrhizin	b	[23]
**58**	rhaoglucoglycyrrhizin	b	[23]
**59**	rhaogalactoglycyrrhizin	b	[23]
**60**	11-deoxo-20α-glycyrrhizin	b	[23]
**61**	20α-galacturonoylglycyrrhizin	b	[23]
**62**	20α-rhaoglycyrrhizin	b	[23]
**63**	glyuralsaponin A	a	[32]
**64**	glyuralsaponin B	a	[32]
**65**	glyuralsaponin C	a	[32]
**66**	glyuralsaponin D	a	[32]
**67**	glyuralsaponin E	a	[32]
**68**	glyuralsaponin F	a	[32]
**69**	glyuralsaponin G	a	[32]
**70**	glyuralsaponin H	a	[32]
**71**	glabasaponin A	b	[36]
**72**	glabasaponin B	b	[36]
**73**	glabasaponin C	b	[36]
**74**	glabasaponin D	b	[36]
**75**	glabasaponin E	b	[36]
**76**	glabasaponin F	b	[36]
**77**	glabasaponin G	b	[36]

a, G. uralensis Fisch.; b, G. glabra L.; c, G. inflate Bat.

**Table 2 molecules-25-03904-t002:** Summary of the biological activities conducted with triterpene saponins in glycyrrhiza ^1^.

No.	Compound	Activity			References
		Property	Method	Major Findings	
**1**	glycyrrhizin (glycyrrhizic acid, uralsaponin A or 18β-glycyrrhizic acid)	Hepatoprotective activities	In vitro—primary rat hepatocytes injured by d-galactosamine (d-GalN)	Lower alanine aminotransaminase (ALT) and aspartate aminotransaminase (AST) levels	[20]
			PLA_2_ inhibitory potency	IC_50_ = 9.3 μM	[20]
			In vitro—primary cultured rat hepatocytes induced by CCl_4_	Prevent soluble enzyme release	[39]
			In vitro—PLC/PRF/5 cells	Modify the expression of hepatitis B virus (HBV)-related antigens on the hepatocytes and suppress sialylation of HBsAg	[40]
			In vivo—BALB/c mice	Suppress increases in AST and ALT, inhibit inducible nitric oxide synthase (iNOS) mRNA expression, and reduce protein and cell infiltration and the degeneration of hepatocytes	[41]
			In vivo—ICR mice	Alleviate CCl_4_-induced liver injury	[42]
			In vivo—Sprague Dawley rats	Exhibit protective effect on retrorsine-induced liver damage	[43]
			In vivo—Wistar rats	Provide partial protection of the liver against ischemia-reperfusion damage	[44]
			In vivo—Wistar rats	Protect against NTiO_2_-induced hepatotoxicity	[45]
		Anti-inflammatory activities	In vitro—lipopolysaccharide (LPS)-stimulated mouse endometrial epithelial cells (MEEC)	Inhibit LPS-induced inflammatory response by inhibiting TLR4 signaling pathway	[47]
			In vitro—neutrophil	Inhibit reactive oxygen species (ROS) generation by neutrophils	[48]
			In vivo—Sprague Dawley rats	Inhibit HMGB1 expression and subsequent production of inflammatory cytokines to prevent cerebral vasospasm (CVS) following subarachnoid hemorrhage (SAH)	[49]
			In vivo—SD rats	Alleviate brain injury after diffuse axonal injury (DAI) via its anti-inflammatory effects	[50]
		Antimicrobial and antiviral activities	In vitro	Inhibit varicella zoster virus (VZV)	[53]
			In vitro	Inhibit severe acute respiratory syndrome coronavirus (SARS-CoV) replication	[54]
			In vitro	Inhibited influenza A virus (IAV) uptake into the cell	[55]
			In vitro	Reduce the severity of an infection with COVID-19 at the two stages of the COVID-19 induced disease process, 1. To block the number of entry points and 2. provide an ACE2 independent anti-inflammatory mechanism.	[57]
			The commercial NA inhibitory screening kit	Possess moderate influenza NA inhibitory activity	[31]
		Cytotoxic and antitumor activities	In vivo—tumor-bearing mice	Reduce expression of TxAS, as well as proliferating cell nuclear antigen (PCNA), and rescue liver and kidney damage	[58]
			In vitro—HepG2	Display protective effects against Aflatoxin B1 (AFB_1_)-induced cytotoxicity	[59]
		Other activities	-	1. Possess immunomodulatory, neuroprotective effects, and antioxidant activities; 2. Bronchitis, peptic ulcers, skin diseases, and oral diseases; 3. Allergic rhinitis	[56,61,62,63,64,65,66]
**3**	licorice-saponin A3	Antimicrobial and antiviral activities	The commercial NA inhibitory screening kit	Possess moderate influenza NA inhibitory activity	[31]
**9**	licorice-saponin G2	Hepatoprotective activities	In vitro—primary rat hepatocytes injured by d-GalN	Lower ALT and AST levels	[20]
			PLA_2_ inhibitory potency	IC_50_ = 16.9 μM	[20]
		Antimicrobial and antiviral activities	The commercial NA inhibitory screening kit	Possess moderate influenza NA inhibitory activity	[31]
**17**	22β-acetoxylglycyrrhizin	Hepatoprotective activities	In vitro—primary rat hepatocytes injured by d-GalN	Lower ALT and AST levels	[20]
			PLA_2_ inhibitory potency	IC_50_ = 27.1 μM	[20]
		Antimicrobial and antiviral activities	In vitro—Madin–Darby canine kidney (MDCK) cells	Inhibit influenza virus A/WSN/33 (H1N1)	[27]
			The commercial NA inhibitory screening kit	Possess moderate influenza NA inhibitory activity	[31]
**21**	uralsaponin D	Hepatoprotective activities	PLA_2_ inhibitory potency	IC_50_ = 32.2 μM	[20]
**25**	licorice-saponin M3(uralsaponin T)	Antimicrobial and antiviral activities	In vitro—MDCK cells	Inhibit influenza virus A/WSN/33 (H1N1)	[27]
			The commercial NA inhibitory screening kit	Possess moderate influenza NA inhibitory activity	[31]
**28–39**	uralsaponins M–Y	Antimicrobial and antiviral activities	In vitro—MDCK cells	Uralsaponin M (28) and uralsaponin S (34) exhibited inhibitory activities against influenza virus A/WSN/33 (H1N1)	[27]
**44–45**	licorice-saponin Q2 (44) macedonoside A (45)	Hepatoprotective activities	In vitro—primary rat hepatocytes injured by d-GalN	Lower ALT and AST levels	[20]
			PLA_2_ inhibitory potency	IC_50_ = 3.6 μM (44) and 6.9 μM (45)	[20]
**63–70**	glyuralsaponins A–H	Hepatoprotective activities	MDA colorimetric assay	Glyuralsaponin B (64) and glyuralsaponin H (70) exhibited moderate antioxidant activities against Fe^2+^/cysteine-induced liver microsomal lipid peroxidation	[32]

^1^, Columns 1 and 2 is the same in Table 1. It should be noted that the empty rows with triterpene saponins are grouped together or not listed in Table 2.

**Table 3 molecules-25-03904-t003:** Summary on solubilization of glycyrrhiza and GL (**1**).

No.	Name	TCM Formulae/TCM/Component	Characteristics	Major Findings	References
**1**	glycyrrhiza	sijunzi decoction, huangqi dazao decoction, baishao gancao decoction	Glycyrrhiza has solubilization effects in three traditional Chinese medicine (TCM) formulae	The solubilizing components in glycyrrhiza are triterpene saponins	[70]
**2**		ephedra	The contents of GL (1), ephedrine, and methephedrine et al. all increase	-	[71]
**3**		epimedium	Icariin in epimedium increases	-	[72]
**4**		curcuma longa	The extractive rate of curcumin double	GL (1) is the main surfactant	[73]
**5**		codonopsis, poria, atractylodes, *Baphicacanthus cusia*, *Paeoniae Radix Alba*, Isatidis Radix, and *Scutellaria baicalensis*	Glycyrrhiza can increase the contents of active ingredients in these TCM	GL (1) is the main surfactant	[74,75,76,77,78]
**6**		*Schisandra chinensis*	No solubilization effects	One of the possible factors affecting the solubilization is some other crude drugs	[70]
**7**	GL (1)	saikosaponin-a	The contents of saikosaponin-a increase	Solubilizing effect is due to GL (1)	[83]
**8**		Ben Lamge granules	The solubility of Ben Lamge granules increases	1. The surface tension of GL (1) decreases; 2. GL (1) exists in micelles in aqueous solution.	[82]
**9**		baicalin	The dissolution rate of baicalin increases	-	[84]
**10**		pueraria	The solubility of pueraria increases	Another possible factor affecting the solubilization is the pH value of the solution	[85]
**11**		pachymic acid	Increase the solubility of pachymic acid	Improve the bioavailability of pachymic acid	[69]
**12**		Puerarin-glycyrrhizic acid dispersible tablets	Improve the dissolution of puerarin	GL (1) possesses solubilization effect	[86]

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
