# Peer review of "Review of Constituents and Biological Activities of Triterpene Saponins from Glycyrrhizae Radix et Rhizoma and Its Solubilization Characteristics"

_molecules, 2020, doi:10.3390/molecules25173904_

Round 1

Reviewer 1 Report

The manuscript provides a review in the constituents of triterpene saponins from Glycyrrhizae Radix et Rhizoma and related bioactive properties. It provides considerable information and it is well written and organized with numerous references to previous studies. For further publication, I would just suggest minor modifications, as follows:

The sentence starting at line 169 (“At present, the world is facing the Corona Virus Disease 2019 (COVID-19) pandemic, caused by severe acute respiratory syndrome coronavirus-2 (SARS-CoV-2). GA (1) could be considered as one of the best constituents which could prove useful against COVID-19.”) should be moved to the conclusions as an example of future works to be considered with these plants, but it remains a speculation since there are no evidences that support its efficacy.

The fact that Figure 1 and 2 have the same title is quite confusing, the authors should consider writing it differently.

In line 209, please substitute “further explaining the” with “further explaining that the”.

Author Response

1. The sentence starting at line 169 (“At present, the world is facing the Corona Virus Disease 2019 (COVID-19) pandemic, caused by severe acute respiratory syndrome coronavirus-2 (SARS-CoV-2). GA (1) could be considered as one of the best constituents which could prove useful against COVID-19.”) should be moved to the conclusions as an example of future works to be considered with these plants, but it remains a speculation since there are no evidences that support its efficacy.

Thank you for your thoughtful suggestions. Surprisingly, in the course of the manuscript revision, we found that there was a new report which could prove that Glycyrrhizin (GL) is useful against COVID-19. So we have revised the sentence “At present, the world is facing the Corona Virus Disease 2019 (COVID-19) pandemic, caused by severe acute respiratory syndrome coronavirus-2 (SARS-CoV-2). GA (1) could be considered as one of the best constituents which could prove useful against COVID-19.” to “At present, the world is facing the Corona Virus Disease 2019 (COVID-19) pandemic, caused by severe acute respiratory syndrome coronavirus-2 (SARS-CoV-2). GL (1) has been used to control COVID-19 infections, which may reduce the severity of an infection with COVID-19 at the two stages of the COVID-19 induced disease process, 1. To block the number of entry points and 2. provide an angiotensin converting enzyme 2 (ACE2) independent anti-inflammatory mechanism.” at line 170 in revised manuscript.

The followed content is for your reference only:

The reference is:

[57] Murck, H. Symptomatic protective action of Glycyrrhizin (Licorice) in COVID-19 Infection? Front Immunol. 2020, 11, 1239. DOI: 10.3389/fimmu.2020.01239.

And the detailed content in the reference is:

Glycyrrhizin (GL) is a frequent component in traditional Chinese medicines, which have been used to control COVID-19 infections. Its systemically active metabolite glycyrrhetinic acid (GA) inhibits 11beta hydroxysteroid dehydrogenase(11betaHSD2) and activates MR in organs, which express this enzyme, including the lungs. Does this affect the protective effect of ACE2? Importantly, GL has anti-inflammatory properties by itself via toll like receptor 4 (TLR4) antagonism and therefore compensates for the reduced protection of the downregulated ACE2. Finally, a direct effect of GL or GA to reduce virus transmission exists, which may involve reduced expression of type 2 transmembrane serine protease (TMPRSS2), which is required for virus uptake. Glycyrrhizin may reduce the severity of an infection with COVID-19 at the two stages of the COVID-19 induced disease process, 1. To block the number of entry points and 2. provide an ACE2 independent anti-inflammatory mechanism.

2. The fact that Figure 1 and 2 have the same title is quite confusing, the authors should consider writing it differently.

Thanks for your detailed suggestions. We have revised “Figure 1. Triterpene saponins in G. uralensis.” to “Figure 1. Triterpene saponins in G. uralensis. (A)” in line 101, and “Figure 2. Triterpene saponins in G. uralensis.” to “Figure 2. Triterpene saponins in G. uralensis. (B)” in line 104.

3. In line 209, please substitute “further explaining the” with “further explaining that the”.

We have revised the “further explaining the” to “further explaining that the” in line 209.

Reviewer 2 Report

The work arises from the clever idea such of the possibility of using these saponins as natural surfactants. From this statement just in the introduction the paper  invites to be read with curiosity. The conclusions are interesting. From my point of view the work has enough potential to be published in a magazine like  Molecules,but it must be subject to a major order review previously.

I suggest the following modifications:

  • Line 56. Add a paragraph describing how the literature review that has been  submittedhas beenperformed : consulted databases, evaluated period,  keywords used for the search. Table 1 could be improved by making the following changes:
    • Eliminate The Columns of Chemical Formula and Molecular Weight, because they do not provide meaningful information
    • Replace the Origin scientific names of species in the Origin Column with numerical codes  1, 2, 3,4,4
    • Include a column called Structure and put in each case the acronym with which the structure appears in Figures 1, 2, 3,
    • Reread the text after having Table 1 modified, and review whether any comments on it  need to be modified, or if any columns can be added to Table 1 so that it is even more informative and better. TRY TO MAKE EVERYTHING SAID IN THE TEXT GRAPHICALLY EXPRESSED IN TABLE 1, IN AN EASY AND QUICK WAY TO UNDERSTAND FOR THE READER.
  1.  
  • Line 117.
    • Add a Table 2, which includes the following columns: 1.No, 2.Triterpenoid, 3.Activity 4. Reference
    • Columns 1 and 2 shall be the same in Table 1. If many rows in Table 2 are empty, they can be grouped together.
    • Column 3 can be subdivided into more columns if the final result is found to be more didactic by breaking down that column.
    • Reread the text after having Table 2, following the same guidelines as specified for Table 1.
    • Lines 201-204, should be  included in the Introduction
  • Line 208.Table 3 should be included where the contents of paragraphs 3.1 and 3.2 are summarized.

Author Response

1. Line 56. Add a paragraph describing how the literature review that has been submitted has been performed: consulted databases, evaluated period, keywords used for the search. Table 1 could be improved by making the following changes:

Eliminate The Columns of Chemical Formula and Molecular Weight, because they do not provide meaningful information

Replace the Origin scientific names of species in the Origin Column with numerical codes 1, 2, 3, 4

Include a column called Structure and put in each case the acronym with which the structure appears in Figures 1, 2, 3, 4

Reread the text after having Table 1 modified, and review whether any comments on it need to be modified, or if any columns can be added to Table 1 so that it is even more informative and better. TRY TO MAKE EVERYTHING SAID IN THE TEXT GRAPHICALLY EXPRESSED IN TABLE 1, IN AN EASY AND QUICK WAY TO UNDERSTAND FOR THE READER.

Thanks for your significant suggestions, we have added a paragraph describing “SciFinder, PubMed, Web of Science, China Journal Net, and relevant English and Chinese literature were used as information sources by the inclusion of the primary search terms ‘Glycyrrhizae Radix’, ‘glycyrrhiza’, ‘Gan-Cao’, ‘liquorice’, ‘licorice’, ‘triterpene saponins’, ‘constituent’, ‘glycyrrhizin’, ‘glycyrrhizic acid’, ‘activities’, ‘solubilization’, and their combinations, mainly from 1984 to 2020.” in line 57 in the revised manuscript according to your modification suggestion.

Ö Table 1 has been revised including eliminating The Columns of Chemical Formula and Molecular Weight, replacing the Origin scientific names of species in the Origin Column with the little English letters a, b, c, instead of the numerical codes 1, 2, 3, 4, because they had been used in the serial number of Table 1.

After days of reflection, we didn’t insert a column called Structure because a lot of the acronym with the structure are the same. We are so sorry that we haven’t thought of a good way to express structure in Figures 1, 2, 3, 4 so far. Then Table 1 has been revised as follows:

No.

Compound

Origin

References

1

glycyrrhizin (glycyrrhizic acid, uralsaponin A or 18β-glycyrrhizic acid)

a, b, c

16 17 18 19 20

2

uralsaponin B

a, b

16 21

3

licorice-saponin A3

a, c

20 22

4

licorice-saponin B2

a, b

22 23

5

licorice-saponin C2

a, b

22 23

6

licorice-saponin D3

a

22

7

licorice-saponin E2

a, c

20 22

8

licorice-saponin F3

a

24

9

licorice-saponin G2

a, b, c

20 23 24

10

licorice-saponin H2

a, b

23 24

11

licorice-saponin J2

a, b

23 24

12

licorice-saponin K2

a, b

21 24

13

licorice-saponin L3

a

25

14

18α-glycyrrhizic acid

a, b

17

15

apioglycyrrhizin

b, c

23 26

16

araboglycyrrhizin

a, b, c

23 26 27

17

22β-acetoxylglycyrrhizin

a, c

20 28

18

3β-O-[β-D-glucuronopyranosyl-

(1→2)-β-D-glucuronopyranosyl]-glycyrretol

a

28

19

3β-O-[β-D-glucuronopyranosyl-(1→2)-β-D-glucuronopyranosyl]-

olean-9,12-diene-30-oic acid

a

28

20

uralsaponin C

a

29

21

uralsaponin D

a, c

20 29

22

uralsaponin E

a

29

23

uralsaponin F

a

29

24

3-O-[β-D-glucuronopyranosyl-

(1→2)-β-D-galactopyranosyl]glycyrrhetic acid

a

30

25

licorice-saponin M3

(uralsaponin T)

a, b

27 31

26

licorice-saponin N4

a, b

31 32

27

licorice-saponin O4

b

31

28

uralsaponin M

a

27

29

uralsaponin N

a

27

30

uralsaponin O

a

27

31

uralsaponin P

a

27

32

uralsaponin Q

a

27

33

uralsaponin R

a

27

34

uralsaponin S

a

27

35

uralsaponin U

a

27

36

uralsaponin V

a, b

21 27

37

uralsaponin W

a

27

38

uralsaponin X

a

27

39

uralsaponin Y

a

27

40

22β-acetoxyl-glycyrrhaldehyde

a, c

20 33

41

3-O-β-D-glucuronopyranosyl-glycyrrhetinic acid

a, b

21 27

42

3-O-[β-D-(6-methyl)glucuro-nopyranosyl (1→2)-D-glucurono-pyranosyl]-24-hydroxyglabrolide

a

34

43

licorice-saponin P2

c

20

44

licorice-saponin Q2

c

20

45

macedonoside A

b, c

20 21

46

24-hydroxy-licorice-saponin E2

c

20

47

macedonoside E

a

35

48

22β-acetyl-uralsaponin C

a

35

49

licorice saponin M1

b

21

50

licorice saponin M2

b

21

51

licorice saponin M3

b

21

52

licorice saponin M4

b

21

53

30-hydroxyglycyrrhizin

b

23

54

glycyrrhizin-20-methanoate

b

23

55

24-hydroxyglucoglycyrrhizin

b

23

56

rhaoglycyrrhizin

b

23

57

11-deoxorhaoglycyrrhizin

b

23

58

rhaoglucoglycyrrhizin

b

23

59

rhaogalactoglycyrrhizin

b

23

60

11-deoxo-20α-

glycyrrhizin

b

23

61

20α-galacturonoylglycyrrhizin

b

23

62

20α-rhaoglycyrrhizin

b

23

63

glyuralsaponin A

a

32

64

glyuralsaponin B

a

32

65

glyuralsaponin C

a

32

66

glyuralsaponin D

a

32

67

glyuralsaponin E

a

32

68

glyuralsaponin F

a

32

69

glyuralsaponin G

a

32

70

glyuralsaponin H

a

32

71

glabasaponin A

b

36

72

glabasaponin B

b

36

73

glabasaponin C

b

36

74

glabasaponin D

b

36

75

glabasaponin E

b

36

76

glabasaponin F

b

36

77

glabasaponin G

b

36

a, G. uralensis Fisch.; b, G. glabra L.; c, G. inflate Bat.

We have carefully read the text to modify it as following:

(1) The sentence “Notably, some glabrolides (7, 21, 22, 30, 39, 42, and 46) were found in G. uralensis and G. inflate,” has been revised to “Notably, some glabrolides (7, 21, 22, 30, 39, 42, and 46) were found in G. uralensis or/and G. inflate,”

(2) The structure of compound 27 in Figure 1 has been deleted.

(3) The structure of compound 8 in Figure 2 has been added.

(4) The structure of compound 1 in Figure 3 has been added.

2. Line 117.

Add a Table 2, which includes the following columns: 1. No, 2. Triterpenoid, 3. Activity 4. Reference

Columns 1 and 2 shall be the same in Table 1. If many rows in Table 2 are empty, they can be grouped together.

Column 3 can be subdivided into more columns if the final result is found to be more didactic by breaking down that column.

Reread the text after having Table 2, following the same guidelines as specified for Table 1.

Thank you for your thoughtful detailed suggestions. According to your modification suggestion, we had added Table 2 in line 142 in revised manuscript and modified seriously the text. The modified text has been used the "Track Changes" function in Microsoft Word. Table 2 has been added as follows:

Table 2. Summary of the biological activities conducted with triterpene saponins in glycyrrhiza 1

No.

Compound

Activity

References

Property

Method

Major findings

1

glycyrrhizin

(glycyrrhizic acid,

uralsaponin A or 18β-glycyrrhizic acid)

Hepatoprotective activities

In vitro - primary rat hepatocytes injured by D-galactosamine (D-GalN)

Lower ALT and AST levels

20

PLA2 inhibitory potency

IC50 = 9.3 μM

20

In vitro - primary cultured rat hepatocytes induced by CCl4

Prevent soluble enzyme release

39

In vitro - PLC/PRF/5 cells

Modify the expression of HBV-related antigens on the hepatocytes and suppress sialylation of HBsAg

40

In vivo - BALB/c mice

Suppress increases in AST and ALT, inhibite iNOS mRNA expression, and reduce protein and cell infiltration and the degeneration of hepatocytes

41

In vivo - ICR mice

Alleviate CCl4-induced liver injury

42

In vivo - Sprague Dawley rats

Exhibit protective effect on retrorsine-induced liver damage

43

In vivo - Wistar rats

Provide partial protection of the liver against ischemia-reperfusion damage

44

In vivo - Wistar rats

Protect against NTiO2-induced hepatotoxicity

45

Anti-inflammatory activities

In vitro - LPS-stimulated MEEC

Inhibit LPS-induced inflammatory response by inhibiting TLR4 signaling pathway

47

In vitro - neutrophil

Inhibit ROS generation by neutrophils

48

In vivo - Sprague Dawley rats

Inhibit HMGB1 expression and subsequent production of inflammatory cytokines to prevent CVS following SAH

49

In vivo – SD rats

Alleviate brain injury after DAI via its anti-inflammatory effects

50

Antimicrobial and antiviral activities

In vitro

Inhibit VZV

53

In vitro

Inhibit SARS-CoV replication

54

In vitro

Inhibited IAV uptake into the cell

55

In vitro

Reduce the severity of an infection with COVID-19 at the two stages of the COVID-19 induced disease process, 1. To block the number of entry points and 2. provide an ACE2 independent anti-inflammatory mechanism.

57

The commercial NA inhibitory screening kit

Possess moderate influenza NA inhibitory activity

31

Cytotoxic and antitumor activities

In vivo –tumor-bearing mice

Reduce expression of TxAS, as well as PCNA, and rescue liver and kidney damage

58

In vitro- HepG2

Display protective effects against AFB1 induced cytotoxicity

59

Other activities

-

1. Possess immunomodulatory, neuroprotective effects, and antioxidant activities; 2. Bronchitis, peptic ulcers, skin diseases, and oral diseases; 3. Allergic rhinitis

56, 61-66

3

licorice-saponin A3

Antimicrobial and antiviral activities

The commercial NA inhibitory screening kit

Possess moderate influenza NA inhibitory activity

31

9

licorice-saponin G2

Hepatoprotective activities

In vitro - primary rat hepatocytes injured by D-galactosamine (D-GalN)

Lower ALT and AST levels

20

PLA2 inhibitory potency

IC50 = 16.9 μM

20

Antimicrobial and antiviral activities

The commercial NA inhibitory screening kit

Possess moderate influenza NA inhibitory activity

31

17

22β-acetoxylglycyrrhizin

Hepatoprotective activities

In vitro - primary rat hepatocytes injured by D-galactosamine (D-GalN)

Lower ALT and AST levels

20

PLA2 inhibitory potency

IC50 = 27.1 μM

20

Antimicrobial and antiviral activities

In vitro - MDCK cells

Inhibit influenza virus A/WSN/33 (H1N1)

27

The commercial NA inhibitory screening kit

Possess moderate influenza NA inhibitory activity

31

21

uralsaponin D

Hepatoprotective activities

PLA2 inhibitory potency

IC50 = 32.2 μM

20

25

licorice-saponin M3

(uralsaponin T)

Antimicrobial and antiviral activities

In vitro - MDCK cells

Inhibit influenza virus A/WSN/33 (H1N1)

27

The commercial NA inhibitory screening kit

Possess moderate influenza NA inhibitory activity

31

28-39

uralsaponins M-Y

Antimicrobial and antiviral activities

In vitro - MDCK cells

Uralsaponin M (28) and uralsaponin S (34) exhibited inhibitory activities against influenza virus A/WSN/33 (H1N1)

27

44-45

licorice-saponin Q2 (44)

macedonoside A (45)

Hepatoprotective activities

In vitro - primary rat hepatocytes injured by D-galactosamine (D-GalN)

Lower ALT and AST levels

20

PLA2 inhibitory potency

IC50 = 3.6 μM (44) and 6.9 μM (45)

20

63-70

glyuralsaponins A-H

Hepatoprotective activities

MDA colorimetric assay

Glyuralsaponin B (64) and glyuralsaponin H (70) exhibited moderate antioxidant activities against Fe2+/cysteine-induced liver microsomal lipid peroxidation

32

1, Columns 1 and 2 is the same in table 1. It should be noted that the empty rows with triterpene saponins are grouped together or not listed in Table 2.

3. Lines 201-204, should be included in the Introduction.

We have modified the text in lines 201-204 to “In nature, saponins are distributed in 90 plant families from 500 genera [67]. Some of them have the potential to be used as natural surfactants because they contain the nonpolar sapogenin and water-soluble sidechain [68]. Glycyrrhiza is the most frequently used TCM in TCM formulae, with the function of harmonizing all kinds of TCMs. Researches on the chemistry, pharmacological effects, clinical applications et al of glycyrrhiza has been very extensive in the last few decades. Besides, the saponins from glycyrrhiza have also significant solubilizing effects [69]. Interestingly, the solubilization characteristics of glycyrrhiza and saponins from glycyrrhiza were studied extensively over recent years. This part of review will deal with the solubilization characteristics, influences, and mechanisms regarding glycyrrhiza and triterpene saponins from glycyrrhiza.” in lines 218-226 in the revised manuscript.

4. Line 208. Table 3 should be included where the contents of paragraphs 3.1 and 3.2 are summarized.

Thanks for your significant suggestions. As advised, we had added Table 3 in line 224 in revised manuscript. Table 3 has been added as follows:

Table 3. Summary on solubilization of glycyrrhiza and GL (1)

No.

Name

TCM formulae/ TCM/Component

Characteristics

Major findings

References

1

glycyrrhiza

sijunzi decoction, huangqi dazao decoction, baishao gancao decoction

Glycyrrhiza has solubilization effects in three TCM formulae

The solubilizing components in glycyrrhiza are triterpene saponins

70

2

ephedra

The contents of GL (1), ephedrine, and methephedrine et al all increase

-

71

3

epimedium

Icariin in epimedium increases

-

72

4

curcuma longa

The extractive rate of curcumin double

GL (1) is the main surfactant

73

5

codonopsis, poria, atractylodes, Baphicacanthus cusia, Paeoniae Radix Alba, Isatidis Radix, and Scutellaria baicalensis

Glycyrrhiza can increase the contents of active ingredients in these TCM

GL (1) is the main surfactant

74-78

6

Schisandra chinensis

No solubilization effects

One of the possible factors affecting the solubilization is some other crude drugs

70

7

GL (1)

saikosaponin-a

The contents of saikosaponin-a increase

Solubilizing effect is due to GL (1)

83

8

Ben Lamge granules

The solubility of Ben Lamge granules increases

1. The surface tension of GL (1) decreases;

2. GL (1) exists in micelles in aqueous solution.

82

9

baicalin

The dissolution rate of baicalin increases

-

84

10

pueraria

The solubility of pueraria increases

Another possible factors affecting the solubilization is the pH value of the solution

85

11

pachymic acid

Increase the solubility of pachymic acid

Improve the bioavailability of pachymic acid

69

12

puerarin–glycyrrhizic acid dispersible tablets

Improve the dissolution of puerarin

GL (1) possesses solubilization effect

86

Round 2

Reviewer 2 Report

Corrections hace been included, so I agree with the acceptance of the paper.